# High-Resolution DLP 3D Printing for Complex Curved and Thin-Walled Structures at Practical Scale: Archimedes Microscrew

**DOI:** 10.3390/mi16070762

**Published:** 2025-06-29

**Authors:** Chih-Lang Lin, Jun-Ting Liu, Chow-Shing Shin

**Affiliations:** 1Center for General Education, Central Taiwan University of Science and Technology, Taichung City 40601, Taiwan; cllin101943@ctust.edu.tw; 2Department of Automatic Control Engineering, Feng Chia University, Taichung City 407802, Taiwan; 3Department of Mechanical Engineering, National Taiwan University, Taipei City 106319, Taiwan; r03522530@ntu.edu.tw

**Keywords:** projection micro-stereolithography (PμSL), DLP 3D printing, microfluidic component, Archimedes microscrew, photo-polymerization

## Abstract

As three-dimensional (3D) printing becomes increasingly prevalent in microfluidic system fabrication, the demand for high precision has become critical. Among various 3D printing technologies, light-curing-based methods offer superior resolution and are particularly well suited for fabricating microfluidic channels and associated micron-scale components. Two-photon polymerization (TPP), one such method, can achieve ultra-high resolution at the submicron level. However, its severely limited printable volume and high operational costs significantly constrain its practicality for real-world applications. In contrast, digital light processing (DLP) 3D printing provides a more balanced alternative, offering operational convenience, lower cost, and print dimensions that are more compatible with practical microfluidic needs. Despite these advantages, most commercial DLP systems still struggle to fabricate intricate, high-resolution structures—particularly curve, thin-walled, or hollow ones—due to over-curing and interlayer adhesion issues. In this study, we developed a DLP-based projection micro-stereolithography (PμSL) system with a simple optical reconfiguration and fine-tuned its parameters to overcome limitations in printing precise and intricate structures. For demonstration, we selected an Archimedes microscrew as the target structure, as it serves as a key component in microfluidic micromixers. Based on our previous study, the most effective design was selected and fabricated in accordance with practical microfluidic dimensions. The PμSL system developed in this study, along with optimized parameters, provides a reference for applying DLP 3D printing in high-precision microfabrication and advancing microfluidic component development.

## 1. Introduction

Additive manufacturing (AM), or 3D printing, has rapidly expanded beyond prototyping and now plays a vital role in biomedical, electronic, and materials science applications [1,2,3,4,5,6]. Among these applications, the use of 3D printing for fabricating microfluidic chips has attracted growing interest [7,8], where the demand for higher printing resolution and precision has become paramount. In particular, recent developments in 3D micromixers [9,10,11,12,13,14] have provided promising solutions for efficient mixing in microchemical processing, microfluidic analysis, and micro total analysis systems (μ-TAS). In various 3D printing techniques, light-curing-based methods offer superior resolution [15], making them particularly suitable for fabricating precise microfluidic systems and related components. Their complex 3D architectures, which are difficult to fabricate using conventional methods, can instead be realized through 3D printing technologies.

In our previous works [16,17], we designed an Archimedean micromixer and demonstrated that rapid mixing over a short channel length can be achieved. It was fabricated using TPP 3D printing, a light-curing-based technique capable of producing complex microstructures with submicron resolution. Although TPP successfully fabricated highly intricate and mechanically robust 3D micromixers, its limited printable volume and high operational costs severely constrain its practicality for broader micro-fluidic applications. For instance, microfluidic channels often range from tens to hundreds of micrometers in width and extend approximately 1–2 cm in length [18,19,20,21]. Under such conditions, using TPP with a typical resolution of 0.1–0.3 μm [22,23] to fabricate even a small 1 mm^3^ structure can take a prohibitively long time. This limitation in scalability makes it difficult for TPP to be adopted widely in practical settings.

To address these limitations, DLP 3D printing has gradually become a promising mainstream approach for microfluidic device fabrication, owing to advantages such as lower cost, greater accessibility, and printable volumes that are more suitable for practical applications [24,25,26,27]. However, conventional commercial DLP 3D printers are often insufficient for applications that demand higher resolution and enhanced structural fidelity. Their native optical configurations, optimized for macro-scale printing, typically lack the precision required to fabricate microscale features with sharp definition. As such, achieving the resolution required for complex microstructures usually necessitates additional optical modifications, such as altering the projection optics to reduce pixel size and improve image focus across a smaller field of view.

On the other hand, DLP systems face considerable challenges when printing complex geometries, especially curved, thin-walled, or hollow structures. These difficulties often lead to printing failures due to over-curing [28] and insufficient inter-layer adhesion [29], which in turn results in dimensional inaccuracies, the gluing together of features in close proximity, or weak bonding between layers. Such issues become even more pronounced when fabricating structures like Archimedean micromixers, which feature spiral manifolds and closely spaced thin-walled elements that must remain clearly separated to maintain unobstructed internal flow paths [17]. Overcoming these challenges requires meticulous tuning of printing parameters to ensure structural fidelity and functional performance.

In this study, we developed a cost-effective and easily implementable optical reconfiguration of a commercial DLP projector to construct a PμSL system, addressing the challenges of fabricating high-resolution and intricate microstructures without the need for high-magnification optics or specialized optical components. A 3D Archimedes microscrew, previously proposed and validated in our earlier work at the micrometer scale using TPP [17], was redesigned to conform to commonly used microfluidic system scales. Preliminary test printing was first carried out to evaluate the limitations on wall separation, wall thickness, and height. The microscrew geometry was eventually finalized based on the design guidelines established in our previous study and subsequently realized using the PμSL system.

## 2. Materials and Methods

### 2.1. Photo-Curable Resin

The resin used is a mixture of Photomer 4012, Photomer 4017, Photomer 4028, and Photomer 8127. Preliminary tests on different compositions suggested that a weight ratio of 8:1:1:1 yields optimal performance. Omnirad 819 (2 wt%) was added as the photoinitiator. All resin components and the photoinitiator were supplied by IGM Resins (IGM Resins International Trading Taiwan Ltd., Taoyuan, Taiwan). Additionally, Sudan I (0.15 wt%, Sigma-Aldrich, Neihu, Taipei, Taiwan) was added as a UV absorbent to limit the curing depth.

### 2.2. PμSL Printing System

A modified commercial DLP projector (P1500, ACER, Taipei, Taiwan) was modified to generate patterned images for 3D printing via photo-polymerization. The projector’s digital micromirror device (DMD) consists of 1920 × 1080 pixels. The major modification involved inverting and repositioning the projector lens assembly to reduce the projected image size to falls within the commonly used dimensions of microfluidic systems.

The original lens module, optimized for large-area imaging, was disassembled and reconfigured by inverting and shortening the distance between the lens and the DMD. This modification reduced the projection area from the default 115.0 mm × 64.7 mm width to approximately 4.3 mm × 2.4 mm, substantially improving pixel resolution from 59.9 μm to 2.2 μm. This simple optical reconfiguration achieves an approximate 27-fold resolution enhancement without additional complex optics, such as converging systems or objective lenses. As a result, this modified projection-based 3D printing system can be classified as a PμSL system, in which structures are fabricated through layer-by-layer curing of a photocurable resin by the ultraviolet component of the dynamically projected images.

The top surface of the print platform is initially aligned with the resin surface. During printing, the platform is lowered into the resin bath in discrete steps. Following each descending step, the appropriate image is projected. This enables layer-by-layer buildup of the object. A top-down configuration (Figure 1) is employed, which eliminates the need for peeling from a release film. This not only saves the overall print time but also prevents damage to previously cured delicate structures during detachment from the release film, thereby enabling more precise and higher-resolution printing. As a result, the system is particularly suitable for fabricating delicate microscale features. Calibration of the print platform motion indicates that a descending step size of 0.6 μm can be achieved with high reproducibility and accuracy.

### 2.3. Method for Evaluating Printable Features

To evaluate the printing accuracy and limitations of the current system, thin walls with various thicknesses and heights were test-printed. Two test-print designs were employed, as shown in Figure 2. In Figure 2a, six vertical walls are connected by two horizontal ties that pass through each wall, providing structural support to help them stand upright in a stable manner. The horizontal separation *d* and height *h* of the walls were kept constant at 400 μm and 350 μm, respectively, while the wall thickness *t* varied from 10 μm to 50 μm across different prints. In Figure 2b, four pairs of vertical walls are shown with a constant thickness of 10 μm. The horizontal separations between successive pairs of walls are 180, 200, 250, and 300 μm, respectively. A series of wall heights *h*, ranging from 100 to 500 μm, were tested in different prints.

The 3D model of each structure was horizontally sliced into layers with thicknesses ranging from 0.8 to 1.0 μm using Creation Workshop version 1.0.0.41 slicing software. Each sliced layer image was projected onto the resin surface with exposure times ranging from 1.3 to 1.6 s. During printing, the platform was lowered using a precision ball screw-driven stage, with step sizes matched to the slicing thickness. These parameters will be further refined based on the printing outcomes of the subsequent preliminary structural tests to ensure sufficient resolution and structural fidelity. The combination of fine slicing and controlled exposure enables high-resolution fabrication of thin-walled and complex structures.

### 2.4. Design and Printing Strategy of Archimedes Microscrew

A 3D solid model of the Archimedes microscrew is generated by simultaneously rotating and translating a rectangular cross-section along the dotted path shown in Figure 3a, resulting in a geometry that forms the screw surface. Figure 3b illustrates the outcome of one complete turn. Based on this design pattern, different screw turns can be constructed. A rectangular cuboid with a height (*H*)-to-width (*W*) ratio of 0.75 is extracted from the circular-sectioned screw, as shown in Figure 3c. The resulting microscrew cuboid (Figure 3d) is then placed in the rectangular cross-sectional flow manifolds formed by two side walls.

The solid model is then horizontally sliced into multiple layers, and the image of each layer is recorded. The total number of layers corresponds to the overall height of the model divided by the thickness of each printing layer. Figure 4 shows examples of the sliced images at three different heights of the model. These successive sliced images are projected onto the printing stage, which is progressively lowered into the resin in discrete steps.

## 3. Results and Discussions

### 3.1. Wall Thickness and Structural Stability

Based on the slicing and exposure settings described previously, preliminary structures were printed using a layer thickness of 1.0 μm and an exposure times of 1.5 s. These parameters were selected to balance curing depth and structural resolution, and were consistently applied throughout the fabrication of the thin-walled structures. As discussed earlier, thinner walls printed under these settings exhibited improved definition but were more susceptible to deformation.

Figure 5 shows scanning electron micrographs (SEMs) of printed structures based on the design in Figure 2a, with wall thicknesses of 10 μm and 20 μm, respectively. In the 10 μm case (Figure 5a), the low stiffness of the thin walls makes them prone to localized buckling, resulting in visible bulging deformation. This phenomenon can be attributed to volumetric contraction associated with polymerization, combined with the layer-by-layer curing process of the current 3D printing technique, which may induce non-uniform residual stress. As the wall thickness increases, its rigidity improves, allowing it to resist buckling more effectively. Consequently, the structure appears nearly normal, and the printed walls become well defined and structurally sound, as shown in Figure 5b.

Beyond deformation, dimensional inaccuracy is also evident. Measurements show that the printed walls are consistently thicker than their corresponding design values. Figure 6 compares the designed and printed wall thicknesses, revealing that the final printed structures are approximately 3.3 times thicker than intended. This discrepancy may arise from three primary factors.

First, the light emitted from each pixel on the DMD does not exhibit a sharply bounded unit function distribution but rather follows a Gaussian profile [30,31]. The light intensity gradually decays and remains non-zero even at a considerable distance beyond the pixel edge. When neighboring pixels are activated simultaneously, such as across a wall’s width, the overlap of beyond-the-pixel light may raise the total intensity sufficiently to initiate unintended polymerization outside the designed boundary.

Second, the resin system used in this work, like most materials employed in stereolithographic 3D printing, cures via a free radical polymerization mechanism [32]. These free radicals can diffuse a short distance from their origins before reacting, contributing to a spatially extended polymerization zone [33].

Third, excessive polymerization may also result from overlapping exposures during the layer-by-layer process. In the current setup, light converges at the focal plane where curing is intended. Below this plane, the beam diverges and becomes weaker due to the inherent divergence and resin absorption. Although the intensity in these regions is insufficient for polymerization during a single exposure, repeated exposures allow free radicals to accumulate. Over time for a number of successive exposures, the radical concentration may exceed the polymerization threshold even in unintended areas, leading to further broadening of the cured walls.

### 3.2. Effect of Wall Height and Gap Spacing on Printable Resolution

Figure 7 shows the printed results based on the design in Figure 2b, with wall heights (h) ranging from 100 to 500 μm and a fixed wall thickness of 10 μm. In each print, the designed horizontal separations between adjacent wall pairs, from left to right, are 180 μm, 200 μm, 250 μm, and 300 μm, respectively. The actual gap widths are smaller than the designed values due to the wall broadening discussed earlier. At a wall height of 100 μm (Figure 7a), the gaps between each wall pair are generally visible, except near the intersections between the horizontal reinforcing ties and the walls. The excessive curing observed there around the corners can be attributed to previously mentioned factors: the superposition of beyond-the-pixel light, short-range diffusion of free radicals, and accumulation of free radicals from successive exposures. Narrow gaps can confine free radicals, facilitating their accumulation. When the wall height increases to 200 μm (Figure 7b), the spaces in the leftmost 180 μm gap and part of the 200 μm gaps are no longer fully open. Interestingly, the localized bulging observed in Figure 5a does not appear here, likely due to the higher stiffness of shorter walls. Vertical wall profiles remain clearly defined, suggesting that the filling occurs only up to a portion of the wall height. The excess material deep within the gaps may be a residual resin that cannot be completely removed due to surface tension in these narrow intervals. As the wall height increases to 300 μm (Figure 7c) and beyond (Figure 7d,e), the aforementioned issues persist. At a wall height of 500 μm, the entire 200 μm gap becomes filled. Moreover, from a height of 300 μm onward, localized wall bulging begins to appear, indicating that wall stiffness is no longer sufficient to withstand buckling caused by the residual curing stress.

The above printing results suggest that to ensure a through flow, the pitch of the Archimedes microscrew must be large enough. Considering the screw profile is much more complicated than a vertical wall, a pitch of 280 μm is chosen to generate the microscrew for a 500 μm channel width. For an 800 μm channel width, the pitch is increased in proportion to 480 μm.

### 3.3. Realization of the Archimedes Micoscrew

According to previous findings, the geometric parameters of a two-turn Archimedes microscrew were carefully selected to ensure structural integrity and to prevent over-polymerization in critical regions. The design of the screw wall was guided by the empirical results shown in Figure 6, where measured deviations between designed and printed thicknesses were used to determine nominal dimensions that would yield a final printed wall thickness of approximately 20 μm. This design strategy was employed to mitigate deformation caused by residual stress, as previously demonstrated. In other words, the results in Figure 6 serve as practical design guidelines for efficiently realizing the proposed structure.

Furthermore, to prevent unintended closure of flow gaps resulting from optical broadening and radical diffusion, the screw pitch was optimized to ensure sufficient spacing between adjacent wall segments in accordance with earlier guidelines. To ensure clear separation between neighboring screw threads and sufficient structural stiffness, a pitch of 280 μm and 480 μm was adopted for the 500 μm- and 800 μm-wide flow channel widths, respectively. The designed microscrew profiles were then fabricated using the same PμSL printing system described earlier.

To evaluate the fabrication results of the printed microscrews, key geometric features were measured from SEM images. Figure 8 presents SEM images of the printed Archimedes microscrew structures with side walls. The average wall thickness was approximately 20.2 ± 0.5 μm, and the pitch spacing was 282.8 ± 5.6 μm and 483.2 ± 5.2 μm for the 500 μm- and 800 μm-wide flow channel widths, respectively. Overall, the screw elements were well resolved, with no significant distortion or merging between adjacent threads, indicating that the chosen geometric parameters successfully mitigated the effects of optical overexposure and radical accumulation. Furthermore, structural fidelity and continuity of the microscrew manifold were achieved, which are expected to facilitate sufficient flow and provide a foundation for future mixing applications.

It is also noteworthy that the surfaces of the screw threads exhibited wave-like striations along the axial direction. These patterns are characteristic of layer-by-layer AM and are generally unavoidable due to the vertical resolution limits of the projection system. Owing to the confined and curved geometry of the spiral structure, direct measurement of surface roughness on these internal walls is not feasible using conventional methods. Nevertheless, such surface irregularities may in fact be beneficial in the context of microfluidic mixing. The undulating features can introduce localized flow perturbations and secondary vortices, which are known to enhance mixing efficiency by disrupting laminar flow patterns. Therefore, these axial striations are considered a functionally favorable byproduct of the fabrication process, rather than a defect.

The printing results of the Archimedes microscrew validate the effectiveness of the design strategy guided by resolution and stability analyses. The investigation of printing parameters demonstrates the feasibility of fabricating complex 3D flow structures using our PμSL system, providing a foundation for the further development of high-performance microfluidic mixing devices.

## 4. Conclusions

In this study, a commercial DLP projector was successfully modified by inverting its lens assembly, without additional complex optical paths or converging lenses, thereby enabling high-resolution 3D printing and constituting a DLP-based PμSL system. We investigated the limitations and capabilities of the modified system for fabricating microstructures, with a focus on the realization of complex curved and helical thin-walled structures—specifically, the Archimedes microscrew. Through systematic evaluations of printed wall thickness, height, and gap resolution, we identified key parameters that govern structural fidelity and dimensional accuracy. The printable gap between walls becomes narrower than designed, especially as the wall height increases. This deviation is attributed to a combination of optical spreading beyond pixel boundaries, free radical diffusion, and cumulative polymerization effects from layer-to-layer exposures. Based on these insights, Archimedes microscrews were successfully designed and printed using optimized geometric parameters. The fabricated structures demonstrate clear separation between adjacent screw threads without structural fusion or collapse, confirming the feasibility of using this printing strategy for complex 3D microfluidic devices. The findings in this study not only offer practical guidelines for printing resolution-sensitive microstructures but also highlight the potential of DLP-based 3D printing in advancing the design and fabrication of integrated microfluidic components.

## Figures and Tables

**Figure 1 micromachines-16-00762-f001:**
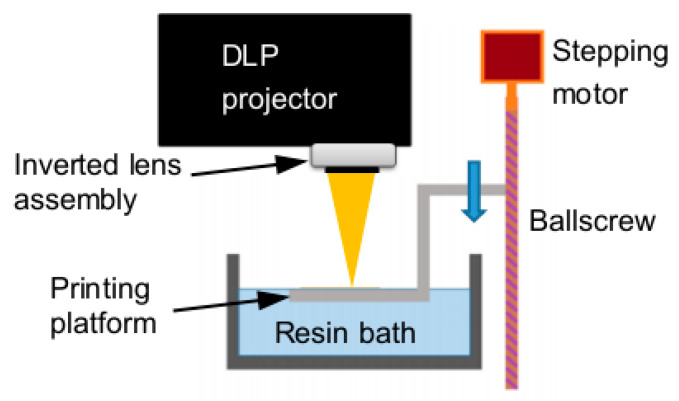
Schematic diagram of the PμSL 3D printing system setup.

**Figure 2 micromachines-16-00762-f002:**
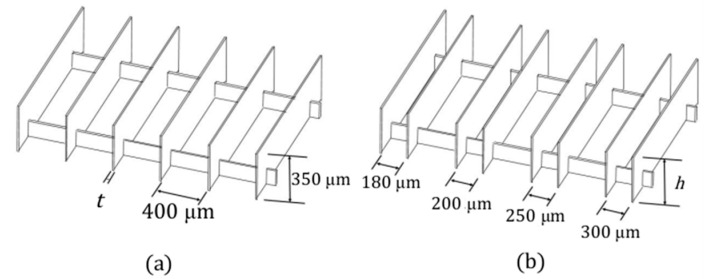
Test print designs for testing (**a**) wall thickness *t* (=10 μm to 50 μm) and (**b**) wall height *h* (=100 μm to 500 μm).

**Figure 3 micromachines-16-00762-f003:**
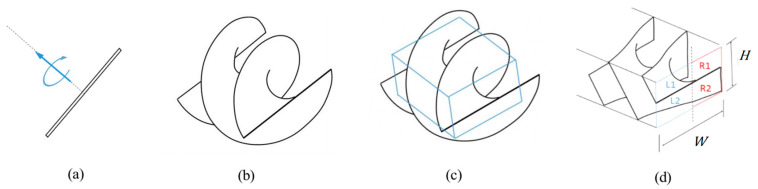
Generation of the Archimedes microscrew model for printing: (**a**) rectangular generator undergoing simultaneous rotation and translation; (**b**) resulting screw surface after one full turn; (**c**) rectangular cuboid section extracted for use; (**d**) final microscrew with rectangular cross-sectional throughput.

**Figure 4 micromachines-16-00762-f004:**
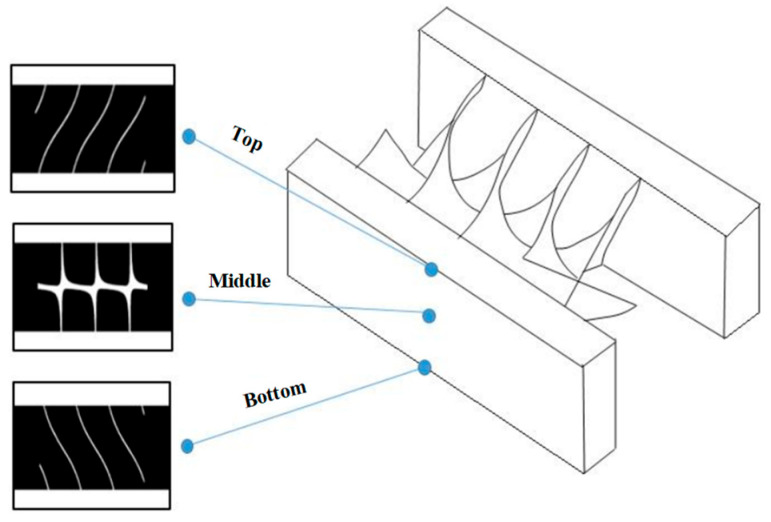
Example of some of the slices of the solid model at three different depths.

**Figure 5 micromachines-16-00762-f005:**
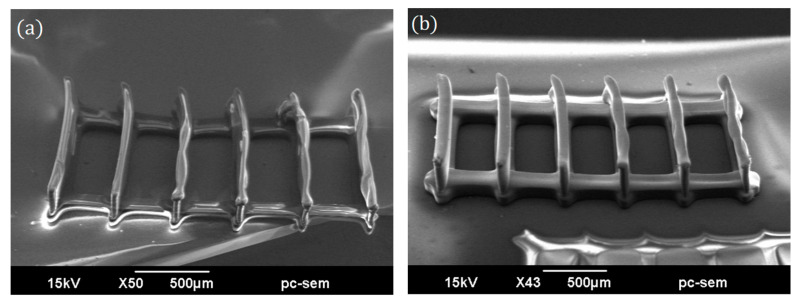
Printed results of the design shown in Figure 2a with designed wall thicknesses of (**a**) 10 μm and (**b**) 20 μm.

**Figure 6 micromachines-16-00762-f006:**
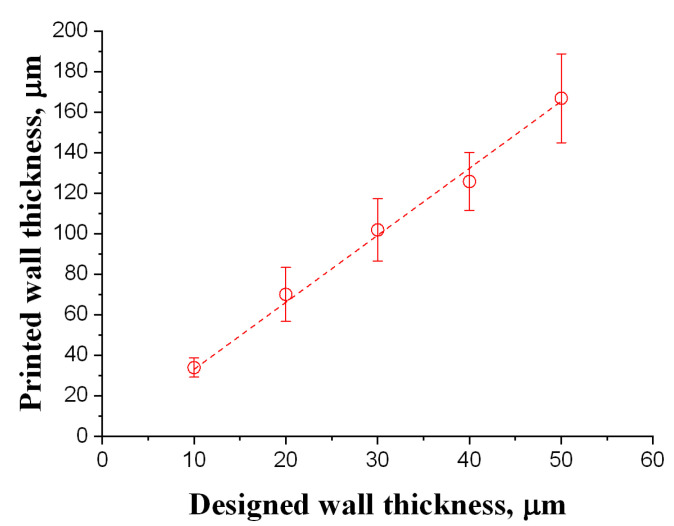
Comparison of the designed and printed wall thicknesses.

**Figure 7 micromachines-16-00762-f007:**
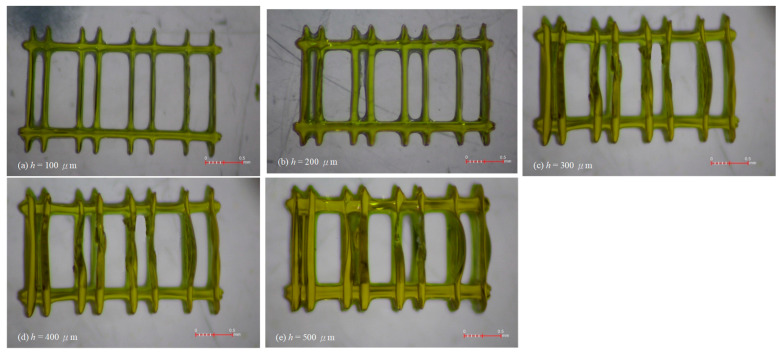
Print results of the design in Figure 2b with a designed wall thickness of 10 μm and a height of (**a**) 100 μm, (**b**) 200 μm, (**c**) 300 μm, (**d**) 400 μm, and (**e**) 500 μm.

**Figure 8 micromachines-16-00762-f008:**
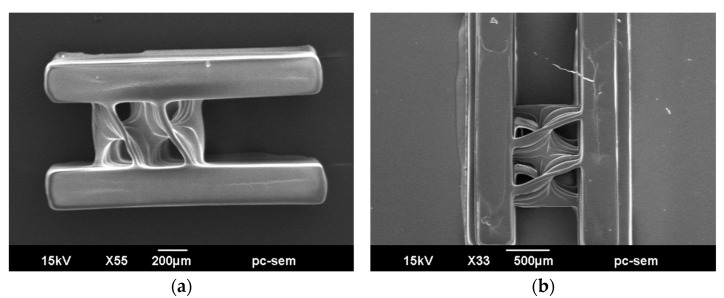
SEM images of the printed Archimedean micromixer structures: (**a**) 500 μm width and (**b**) 800 μm width.

## Data Availability

The data involved in this study have basically been presented in this paper. Further details and queries may be directed at the corresponding author.

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
