# Peer review of "High-Resolution DLP 3D Printing for Complex Curved and Thin-Walled Structures at Practical Scale: Archimedes Microscrew"

_micromachines, 2025, doi:10.3390/mi16070762_

Round 1

Reviewer 1 Report

Comments and Suggestions for Authors

This study presents the modification a DLP-based photopolymerization 3D printing system, aiming to address challenges related to resolution and structural integrity when printing complex curved and microscale structures. The successful fabrication of an Archimedes microscrew demonstrates the system’s practical potential for microfluidic device manufacturing. Overall, the manuscript is well-structured and data-rich. However, several critical details remain insufficiently described, and the use of technical terminology lacks consistency.

The following issues should be carefully considered and revised:

  1. The implementation of this work is based on modifications to the experimental equipment. However, the optical design rationale and technical implementation details of these modifications are not sufficiently explained. For instance:
  • Which specific optical components were changed, replaced, or reconfigured (e.g., focal length, lens distance, numerical aperture)?
  • How does the inversion and repositioning of the lens assembly reduce the projection area while maintaining resolution and image quality?
    A clearer explanation of the optical path and how it contributes to printing resolution is essential.
  1. In Section 2.3, the authors describe the evaluation method for printable features but do not provide any details about the slicing strategy or key printing parameters (such as exposure time, layer thickness, light intensity, or printing speed). These parameters critically influence the dimensional accuracy and resolution of printed microstructures.
  2. Figure 6 uses bar charts to compare the designed wall thickness with the actual printed thickness. However, the text clearly states that the printed walls are generally three to four times thicker than designed, suggesting a point-wise linear relationship. Bar charts are not suitable for representing this kind of linear correlation.
  3. The manuscript contains inconsistencies in the definition and usage of abbreviations. Professional terms should be clearly defined upon their first appearance and consistently abbreviated thereafter. Examples include: (1) Three-dimensional (3D), but the full term is repeatedly used again in lines 44 and 76 without abbreviation. (2) The term “digital micromirror device (DMD)” is written in full in both line 95 and line 170, without consistent use of the abbreviation. It should be defined once upon first appearance and abbreviated as “DMD” thereafter throughout the manuscript.

Author Response

Comments 1: 

  1. The implementation of this work is based on modifications to the experimental equipment. However, the optical design rationale and technical implementation details of these modifications are not sufficiently explained. For instance:
  • Which specific optical components were changed, replaced, or reconfigured (e.g., focal length, lens distance, numerical aperture)?
  • How does the inversion and repositioning of the lens assembly reduce the projection area while maintaining resolution and image quality? A clearer explanation of the optical path and how it contributes to printing resolution is essential.

Responses 1:

We appreciate the reviewer’s insightful comment regarding the optical design. To address this point, we have added a more detailed explanation of the modifications made to the optical setup and the rationale behind them in the revised manuscript, specifically in Section 2.2 (PμSL Printing System) and the Abstract. A brief summary is provided below for clarity.

The major modification involved inverting and repositioning the projector lens assembly to reduce the projected image size. The original lens module, optimized for large-area imaging, was disassembled and reconfigured by inverting and shortening the distance between the lens and the DMD. As a result, this modification reduced the projection area from the default 115.0 mm × 64.7 width to approximately 4.3 mm × 2.4 mm, substantially improving pixel resolution from 59.9 μm to 2.2 μm. This simple optical reconfiguration achieves an approximate 27-fold resolution enhancement without additional complex optics, such as converging systems or objective lenses.

Comments 2:

In Section 2.3, the authors describe the evaluation method for printable features but do not provide any details about the slicing strategy or key printing parameters (such as exposure time, layer thickness, light intensity, or printing speed). These parameters critically influence the dimensional accuracy and resolution of printed microstructures.

Responses 2:

We thank the reviewer for pointing out the lack of printing parameter details. In response, we have added the slicing strategy and relevant printing parameters to Section 2.3 (Method for evaluating printable features) and Section 3.1 (Wall thickness and Structural Stability) of the revised manuscript. A brief summary is provided below for clarity.

The printing parameters are crucial for addressing the challenges of fabricating high-resolution and intricate microstructures. Specifically, slicing layer thicknesses ranging from 0.8 to 1.0 μm were tested, and the printing stage was actuated using a precision ballscrew-driven platform, with step sizes matched to the slicing thickness. Exposure times ranging from 1.3 to 1.6 seconds were also evaluated for each layer. These parameters will be further refined based on the printing outcomes of the subsequent preliminary structural tests to ensure optimal resolution and structural fidelity.

Comments 3:

Figure 6 uses bar charts to compare the designed wall thickness with the actual printed thickness. However, the text clearly states that the printed walls are generally three to four times thicker than designed, suggesting a point-wise linear relationship. Bar charts are not suitable for representing this kind of linear correlation.

Responses 3:

We appreciate the reviewer’s valuable suggestion regarding the appropriateness of the data visualization in Figure 6. In response, we have replaced the bar chart with a scatter plot comparing the designed wall thickness with the measured printed thickness. This revised figure more accurately reflects the linear trend discussed in the text and provides a clearer visualization of the proportional relationship between design and outcome. The updated Figure 6 has been included in the revised manuscript accordingly.

Comments 4:

The manuscript contains inconsistencies in the definition and usage of abbreviations. Professional terms should be clearly defined upon their first appearance and consistently abbreviated thereafter. Examples include: (1) Three-dimensional (3D), but the full term is repeatedly used again in lines 44 and 76 without abbreviation. (2) The term “digital micromirror device (DMD)” is written in full in both line 95 and line 170, without consistent use of the abbreviation. It should be defined once upon first appearance and abbreviated as “DMD” thereafter throughout the manuscript.

Responses 4: 

We thank the reviewer for pointing out the inconsistencies in the use and definition of abbreviations throughout the manuscript. In response, we have carefully reviewed the manuscript and revised all instances to ensure consistency. All other abbreviations in the manuscript have also been reviewed to ensure they are defined only once and used consistently thereafter.

Reviewer 2 Report

Comments and Suggestions for Authors

This paper presents the development of a micro-DLP 3D printing system and provides results from a parametric study evaluating the printing quality and stability of the proposed setup. Additionally, the fabrication of an Archimedes microscrew using the system is demonstrated. The overall research design appears appropriate; however, several important issues need to be addressed:

  1. The main contribution of the work should be more clearly highlighted. As micro-DLP is a well-established printing technique, it is essential to emphasize what specific improvements or novel aspects are introduced in the proposed system.

  2. In Fig. 6, there is a noticeable discrepancy between the designed geometry and the fabricated parts. If any compensation techniques were applied to improve fabrication accuracy—particularly for the microscrew—please clarify them. If no compensation was used, an explanation should be provided regarding how a 20-micrometer-thick wall was successfully fabricated.

  3. Section 3.3 includes only qualitative analysis of the fabricated parts. Incorporating quantitative measurements (e.g., dimensional accuracy, surface roughness, or resolution metrics) would significantly enhance the clarity and rigor of the analysis.

Author Response

Comments 1:

The main contribution of the work should be more clearly highlighted. As micro-DLP is a well-established printing technique, it is essential to emphasize what specific improvements or novel aspects are introduced in the proposed system.

Responses 1:

We thank the reviewer for the thoughtful feedback. We agree that micro-DLP is an established technique, and we have now revised the manuscript to more clearly emphasize the specific contributions and novel aspects of our work.

In particular, we highlight the following key improvements:

(1) We propose a cost-effective and easily implementable optical reconfiguration of a commercial DLP projector that significantly enhances printing resolution to 2.2 μm per pixel, without the need for additional high-magnification optics such as objective lenses, and without requiring any specialized optical components or alignment systems.

(2) A parametric investigation was conducted to evaluate how wall thickness, spacing, and height affect print fidelity, providing practical guidelines for resolution-sensitive microfabrication.

(3) As a demonstration of application potential, we fabricated an Archimedes microscrew with complex, curved, and thin-walled features, which are typically challenging for commercial DLP systems. The success of this print illustrates the practical effectiveness of our modified system for intricate 3D microfluidic components.

These contributions are now more explicitly stated in the Abstract, Introduction, and Conclusions sections to help readers and reviewers better appreciate the novelty and value of this study.  

Comments 2:

In Fig. 6, there is a noticeable discrepancy between the designed geometry and the fabricated parts. If any compensation techniques were applied to improve fabrication accuracy—particularly for the microscrew—please clarify them. If no compensation was used, an explanation should be provided regarding how a 20-micrometer-thick wall was successfully fabricated.

Responses 2:

We appreciate the reviewer’s observation regarding the dimensional discrepancy in Fig. 6 and the follow-up question concerning the fabrication accuracy of the microscrew walls.

In our study, the design of the Archimedes microscrew was directly guided by the results of preliminary printing tests shown in Fig. 6. By systematically examining the deviations between the designed and printed wall thicknesses across various geometries, the wall dimensions of the microscrew were chosen such that the resulting printed structures would yield a wall thickness of approximately 20 μm. This design approach avoids additional complexity in post-processing or geometric compensation. In other words, the results in Fig. 6 serve as practical design guidelines for efficiently realizing the proposed structure.

To clarify this point, we have added the following sentence to Section 3.3 of the revised manuscript:

“The design of the microscrew walls was guided by the empirical results shown in Fig. 6, where measured deviations between designed and printed thicknesses were used to determine nominal dimensions that would yield a final printed wall thickness of approximately 20 μm. This design approach ensured sufficient rigidity and structural integrity while maintaining printability and resolution. In other words, the results in Fig. 6 serve as practical design guidelines for efficiently realizing the proposed structure.”

Comments 3:

Section 3.3 includes only qualitative analysis of the fabricated parts. Incorporating quantitative measurements (e.g., dimensional accuracy, surface roughness, or resolution metrics) would significantly enhance the clarity and rigor of the analysis.

Responses 3: 

We appreciate the reviewer’s suggestion. In response, we have added a new paragraph to Section 3.3 providing quantitative measurements derived from SEM analysis, including average wall thickness and pitch spacing deviations of the printed Archimedes microscrew. These results confirm the dimensional fidelity and structural integrity of the fabricated components.

As for surface roughness, we acknowledge that direct quantitative measurement of the interior surfaces of the spiral structures is not feasible due to their confined and curved geometry. Conventional profilometry or AFM techniques are not applicable in such narrow and enclosed features. However, high-magnification SEM images clearly reveal periodic wave-like striations on the screw thread surfaces. These striations are consistent with the layer-by-layer fabrication nature of projection micro-stereolithography (PμSL) and represent a form of inherent surface texturing. While we cannot provide exact numerical roughness values, we have added a discussion in the revised manuscript highlighting that these striated patterns may in fact be beneficial in microfluidic mixing applications. Specifically, the undulating surfaces can introduce additional microscale perturbations in the flow, potentially enhancing mixing efficiency by disrupting laminar flow patterns.